# Titanium Dioxide/Chromium Oxide/Graphene Oxide Doped into Cellulose Acetate for Medical Applications

**DOI:** 10.3390/polym15030485

**Published:** 2023-01-17

**Authors:** Latifah Mohammed Ali Almaieli, Mai M. Khalaf, Mohamed Gouda, Sultan Alhayyani, Manal F. Abou Taleb, Hany M. Abd El-Lateef

**Affiliations:** 1Department of Chemistry, College of Science, King Faisal University, Al-Ahsa 31982, Saudi Arabia; 2Chemistry Department, Faculty of Science, Sohag University, Sohag 82524, Egypt; 3Department of Chemistry, College of Sciences & Arts, King Abdulaziz University, Rabigh 17371, Saudi Arabia; 4Department of Chemistry, College of Science and Humanities, Prince Sattam bin Abdulaziz University, Al-Kharj 11942, Saudi Arabia; 5Department of Polymer Chemistry, National Center for Radiation Research and Technology (NCRRT), Egyptian Atomic Energy Authority, Cairo 117762, Egypt

**Keywords:** Cr_2_O_3_, TiO_2_, GO, cell viability, wound healing

## Abstract

Wound dressings have been designed based on cellulose acetate encapsulated with different concentrations of chromium oxide (Cr_2_O_3_) and titanium oxide (TiO_2_) with/without graphene oxide (GO). This study comprises the structural, morphological, optical, thermal, and biological behavior of chromium oxide/titanium dioxide/graphene oxide-integrated cellulose acetate (CA) films. The CA-based film bond formation was introduced by functional group analysis via Fourier transform infrared (FTIR) spectroscopy. The fabricated Cr_2_O_3_/TiO_2_/GO@CA film SEM micrographs demonstrate transition metal oxides Cr_2_O_3_ and TiO_2_ on a nano-scale. The TiO_2_@CA shows the lowest contact angle with 30°. Optically, the refractive index increases from 1.76 for CA to 2.14 for the TiO_2_@CA film. Moreover, normal lung cells (A138) growth examination in a function of Cr_2_O_3_/TiO_2_/GO@CA film concentration is conducted, introducing 93.46% with the usage of 4.9 µg/mL. The resulting data showed a promising wound-healing behavior of the CA-based films.

## 1. Introduction

Wound healing is a complex process. It occurs in injured skin with progression through various stages, for instance, hemostasis, inflammation, proliferation, and remodeling, according to the intensity and impact of the injury [1]. In recent times, there has been rapid growth in tissue engineering research for medical purposes which is supported by existing traditional transplant techniques. The main goal of tissue engineering is to discover an appropriate strategy for bio-scaffold formation and optimization to perform as a natural extracellular matrix [2]. The anti-inflammatory features of the designed bio-material are crucial, for instance, biocompatibility and biodegradability. Additionally, a tailored surface for physical and chemical adhesion is considered an important factor [3,4,5,6]. As well, several polymers were tested as basic constituents for bio-scaffold applications due to their biocompatibility.

Among them, cellulose (C_6_H_10_O_5_)_n_ is considered an accessible polymer that occurs naturally. The three active OH function groups per molecule introduce appropriate physicochemical characteristics to be connected with other substances such as graphene oxide, which can facilitate their reinforcement behavior [7]. These properties might suggest the combination of cellulose as a simple strategy to improve its mechanical properties to be appropriate for medical applications. Biodegradable biopolymers are an eco-friendly alternative; in addition, they originate from renewable and biodegradable sources. Additionally, CA is a biodegradable thermoplastic polymer obtained by cellulose esterification, which is the most plentiful semi-synthetic polysaccharide in the surrounding environment [8]. In addition, NPs merging with polymeric constituents improves their bio-applicability [7]. With further in-depth studies on the mechanisms of metals in wound healing, wounded skin restoration methods based on metals have attracted more attention. The amazing biological activity of metal-based materials embraces the antibacterial action of the wound remediation process.

Regarding transition metal oxides, chromium oxide/Cr (III) is considered a vital, multi-functional material [9,10]. Undeniably, the performance of an inorganic material as a germicidal agent is more appropriate than an organic one because of its stability under different conditions. Generally, trioxide metals inhibit the bacterial resistance phenomenon by releasing oxonium ions (H_3_O^+^). Oxonium ions alter pH, causing failure in bacterial growth [11]. In addition, chromium oxide (Cr_2_O_3_) displays antibacterial, antifungal, antioxidant, and antiviral performance [10]. On the other hand, titanium dioxide (TiO_2_) is non-combustible, low soluble, and thermally stable, and it is not categorized as hazardous according to the United Nations (UN) Globally Harmonized System of Classification and Labeling of Chemicals (GHS). Nano-scale titanium dioxide displays three different phases equivalent to the diverse temperature, comprising anatase, rutile, and brookite. Among these phases, anatase has been demonstrated to have highly attractive features for numerous industrial usages, for instance, photo-catalysis, chemical sensors, microelectronics, and self-cleaning and anti-bacterial products. The roles of both Cr_2_O_3_ and TiO_2_ have interfered. Both of them were selected due to their high optical properties, which are appropriate to support the optical properties of the designed scaffolds. Further, GO nano-sheets were suggested to act as filler sheets to gather the metallic oxides and to support the stability of the obtained films. CA, considered a good candidate for numerous medical applications with its properties of high biocompatibility and chemical stability, is the carrier of these oxides to be delivered to the injured position to accelerate the healing procedure.

Further, it has been proved that a trace amount of GO leads to significant modification, while higher doses are toxic. Additionally, GO addition endorses adhesion and proliferation potential [2]. Biologically, GO biocompatibility [12,13], besides its surface active functional groups, offers hydrophilic behavior [14]. Selvi et al., 2016, discuss the heightened physical and chemical characteristics, and biological responses of NC tactics in the biomaterial fabrication process [15]. Prakash et al., 2021, reported the CA/GO/TiO_2_/curcumin NC as an antibacterial agent against wound pathogens. Additionally, it enhances swelling and decomposition behavior which boosts wound healing [1]. Khalid et al., 2017, introduce a successful integration between TiO_2_ NPs and cellulose. The NC bandage displayed good healing with 71% wound contraction [16]. Prakash et al., 2020, have reported the GO/TiO_2_ NC as an antibacterial agent in the presence of light [17]. This is explained by the photocatalytic activity of NCs, besides the release of ROS in light. Moreover, the GO/TiO_2_ NC displays clear biocompatible and hemocompatibility, which boosts wound healing capability [17]. In this work, chromium oxide (Cr_2_O_3_), titanium dioxide (TiO_2_), graphene oxide (GO), and cellulose acetate (CA) will be unified to obtain a series of NC films with type/amount variations in raw oxy-compositions. The physical behaviors, including the structural, morphological, and optical performance of the obtained films, will be studied, as well as the cellular response to the human lung cell line in vitro.

## 2. Materials and Methods

### 2.1. Materials

Chromium (III), titanium oxides, and cellulose acetate with an average molecular weight of 30,000 g/mol), graphite, and other chemicals were purchased from Sigma-Aldrich Co., Burlington, NJ, USA.

### 2.2. Preparation of Scaffold with Different Contents of Oxides

GO is prepared in the laboratory by a modified Hummers’ method [18,19]. Moreover, the CA films were formed with the casting methodology. To obtain the fibrogenic solution, CA was dissolved in acetone at the ratio of 1:10 (*m*/*v*) = 10.0 wt.%, with the powder ingredients tabulated in Table 1, resting for 24 h in a sealed glass at 25 °C. Each sample containing 2 g of CA was dissolved in 20 mL of acetone (10.0 wt.%), then 0.25 g of the powdered oxides were added through this quantity of dissolved CA solution. Thus, the ratio of the additional oxides into the CA is around 12.5% and it was maintained for all samples, except the first one which represents the pure CA as a control one. Then, the solution was subjected to magnetic stirring for 2 h. The solution was poured onto a glass plate, left to rest until complete solvent evaporation. The ratio of the additional oxides was marinated at 0.25 g for all samples, except the first one which is pure CA. This ratio was selected after experimental trials. In detail, increasing the additional oxide powder in the polymeric solution might reduce the stability of the film and may deteriorate the coherence of the scaffold. However, the low contribution of the additional powder to the polymeric solution might not be effective, or, in some cases, it does not appear in the characterizations. Therefore, the balance between the appropriate stability of the obtained scaffold and the appearance of the additives through characterization is highly required. Thus, several trials can be carried out to select an appropriate ratio to be used for a number of compositions to be examined for medical applications.

### 2.3. Characterizations

#### 2.3.1. XRD Measurements

X-ray diffraction (XRD) analyses were carried out via the X-ray diffractometer model (Panalytical-X’pert pro carried out using Cu k_α1_ radiation, λ = 1.5404 Å, 45 kV, 40 mA, Malvern, WR14 1XZ, Worcestershire, UK). It was used mainly to investigate the phase composition of the as-synthesized film composite. All XRD curves were scanned in 5° ≤ 2θ ≤ 70° scope with a step size of 0.02° and a step time of 0.5 s.

#### 2.3.2. FTIR Measurements

Fourier-transformed infrared (FTIR) spectra were scanned via an FTIR spectrometer (Perkin-Elmer 2000 Akron, OH, USA) in the range of 4000–400 cm^−1^. The samples were prepared for the FTIR test by combining the films with KBr and a ratio of 20:1. The samples then were fixed on the holder. Finally, the sample was placed in the front of the beam and scanned within the range of 400 to 4000 cm^−1^.

#### 2.3.3. Examination of Films Morphology

FESEM (QUANTA-FEG250, Kolkata, WB, India) was used to investigate the morphology of MG-63 cells on the films-based NC. For this issue, the nano-film was sanitized using a UV lamp for 30 min of exposure. Each sample was cropped into two pieces of 0.5 × 0.5 cm^2^, then they were added to 12-well plates. An amount of 1.5 mL of MG-63 cells was added to each well. The plate was then incubated at 37 °C for 3 days. After this time, the films were washed with phosphate-buffered saline (PBS) (Merck, Beijing, China). To keep the cells fixed on the film surface, the scaffolds were submerged in a glutaraldehyde solution (4% concentration) (Merck, Beijing, China) for 1.0 h. Then, they were dehydrated in the air for ¼ h. Finally, they were coated with gold for just 2 min to be ready for FESEM resultant surface images.

#### 2.3.4. Thermogravimetric Analysis

TGA was carried out from room temperature up to 600 °C in a thermal analyzer (DTG-60H SHIMADZU, Kyoto, Japan) using the airflow rate of 100 mL/min. The heating rate was 10 °C/min.

#### 2.3.5. UV Measurements

Double beam UV–visible (UV–Vis) (Shanghai Metash instruments Co., Shanghai, China) spectroscopy was used to investigate the optical properties of polymeric samples. Each sample was cropped with a dimension of 2 cm × 2 cm. The thickness of these films was maintained at 0.185 mm.

#### 2.3.6. Contact Angle

A piece of 1 cm × 1 cm was obtained for each sample. Then, it was placed in front of a small camera on a plate. A drop of deionized water was dropped on the horizontal film, while the camera is working, to take one image for each drop on the film. The camera used was a “1600× 8 LED, Ningbo, China”. The images were taken manually after 1 s of the drop.

#### 2.3.7. In Vitro Cell Viability Tests

The normal lung cells (A138) were used under culturing conditions in Dulbecco’s modified Eagle’s medium (DMEM, Gibco, ThermoFisher Scientific, Gillingham, UK) to investigate cell viability. Cells with a density of 5 × 10^3^ (cells/cm^2^) were cultured on the films through 24-well plates, then they were incubated at 37 °C. After 3 days of incubation, media was detached, and MTT (3-(4,5-dimethylthiazol-2-yl)-2,5-diphenyltetrazolium bromide)(Merck, Beijing, China) was injected into each well, then cell viability was measured via an optical analyzer (The Vi-Cell XR Cell Viability Analyzer, Beckman Coulter, Indianapolis, IN, USA).

## 3. Results and Discussion

### 3.1. Structural Investigation

The XRD patterns exhibit cellulose acetate-based NC films (Figure 1). The main intense, broad peak at 2θ = 21° refers to the CA composition [17]. Regarding Cr_2_O_3_ peaks upon the Cr_2_O_3_@CA binary composition, the high intense characteristic peaks at 24.1 and 33.2° is attributed to their existence. Concerning the TiO_2_@CA film, high intense peaks of (1 0 1), (0 0 4), (2 0 0), (1 0 5), (2 0 4), and (1 1 6), are located at 2θ = 25.1°, 37.7°, 48.0°, 55.1°, 62.7°, and 68.5°, respectively, which correspond to anatase TiO_2_ compositions [1].

The noticeable resultant peaks are sharp because of the high sintering temperature during merged oxides preparation. Additionally, the cellulose acetate peak decreased in intensity upon binary NC formation that indicated the successful incorporation of transition metal oxides [1]. GO peak is hidden in Cr_2_O_3_/TiO_2_/GO@CA film owing to the interruption of its 2D crystal lattice by inserting the rest of the composition. Thus, the XRD data verified the chemical structure of the studied films.

FTIR analysis is tested in the wave number scope (400 to 4000 cm^−1^). FT-IR analysis of CA, Cr_2_O_3_@CA, TiO_2_@CA, Cr_2_O_3_/TiO_2_@CA, and Cr_2_O_3_/TiO_2_/GO@CA films are given in Figure 2. The CA spectra exhibit bands at 1030, 1230, 1372, and 1745 cm^−1^ equivalent to C–O–C, C–O–C, C–CH_3_, and C–O, respectively. In the binary compositions (transition metal oxide@CA) film, the intensity of CA peaks from 3000 cm^−1^ to 3400 cm^−1^ declines because of Cr_2_O_3_/TiO_2_ incorporations. This peak is attributed to the stretching mode of the OH group, and changes in its appearance reflect the H-bond between CA and metal oxides, which indicates the film’s compatibility [20]. The GO characteristic peaks at 1150 and 1520 cm^−1^ are assigned to the C–O stretching mode [1]. Regarding Cr_2_O_3_ bands_,_ the weak band at 580 cm^−1^ refers to Cr–O stretching [21]. The FTIR peaks also verified the chemical composition of the prepared films.

The quantitative and qualitative data of the Cr_2_O_3_/TiO_2_/GO@CA film are obtained via the EDX technique. The quantitative detection of elements is obtained by the signal intensities [10]. The data in Figure 3 (inset) confirm the existence of C, N, O, Ti, and Cr elements with 58.89, 4.71, 36, 0.17, and 0.22%. Undeniably, oxygen represents the second uppermost atomic percent after carbon because of its appearance with almost all individual elements at 0.5 keV, as shown in Figure 3 [22]. Lately, technologies assembly of EDX with SEM is executed for giving the quantification picture of materials’ chemical composition [10]. EDX analysis affirmed the successful synthesis and the existence of the corresponding elements (Figure 3).

### 3.2. Morphological Investigation

Nano-composites’ topographical and elemental information was displayed in SEM micrographs which appear in Figure 4, where Figure 4a displays micrographs for Cr_2_O_3_@CA, Figure 4b for TiO_2_/Cr_2_O_3_@CA, and Figure 4c for Cr_2_O_3_/TiO_2_/GO@CA. The binary polymeric film Cr_2_O_3_@CA micrograph confirms the good distribution of chromium (III) oxide grains upon the CA matrix. The grains appeared with semi-spherical shaping, recording an average grain size of 0.32 µm. Additionally, TiO_2_/Cr_2_O_3_@CA ternary film shows a significant reduction in the size of Cr_2_O_3_ grains. Moreover, TiO_2_ merging is accompanied by originating of narrow grooves in between metal oxide grains with better distribution over the CA matrix. Further, the Cr_2_O_3_/TiO_2_/GO@CA film demonstrates with resolution scopes in Figure 4c. The oxides grains show less aggregation tendency with better grain distribution over the polymeric matrix. The close scope displays GO sheets with a decent spread of transition metal oxides. Both Cr_2_O_3_ and TiO_2_ appear in the nano-scale (<50 nm). This surface alteration points to expected variations in biological responses. Indeed, surface grooves are vital in biomaterials’ biological potential.

### 3.3. Wettability Study via Contact Angle Detection

Contact angle measuring is performed to appraise the wettability of polymeric films, as shown in Figure 5. Undeniably, the narrower the contact angle, the greater the adhesion potential, and also the higher the bio-applicability. The wettability angles for CA, Cr_2_O_3_@CA, TiO_2_@CA, Cr_2_O_3_/TiO_2_@CA, and Cr_2_O_3_/TiO_2_/GO@CA films are 38, 39, 30, 37, and 41.5, respectively. In brief, the chemical composition directly influences contact angle and adhesion capability.

Iqhrammullah et al., 2021, have studied the wettability of CA–polyurethane film; the values of the contact angle scope start from 38.2° to 62.5°. The contact angle value of the CA was detected to be lower when compared to the CA/polyurethane films [23]. The CA resultant contact angle matches our study’s experimental data. Based on the previous experimental values, the films’ wettability properties tended to have more hydrophilic behavior with the insertion of TiO_2_. The wettability potential recommends the usage of such NC films biologically.

### 3.4. Thermal Study Using a Thermo-Gravimetric Analysis Technique

The thermal behavior of Cr_2_O_3_/TiO_2_/GO@CA has been investigated by following the weight loss % of NC film by raising the temperature, using the thermo-gravimetric analysis technique (TGA). The oxy-nano-composite is exposed to temperature scope (20–600 °C) at a rate of 20 °C/min under nitrogen gas. Figure 6 shows the thermal behavior with the temperature peak of tested NC weight loss. The decomposition exhibits two overlapped steps with temperature peaks of 354.7 and 452 °C. The two steps reveal temperature scopes of (300–380 °C) and (380 to 460 °C). Regarding the TGA thermogram obtained for the FAC-based NC, originally, there is an insignificant mass loss of 7% up to 250 °C, corresponding to the H_2_O attach to the hydrophilic hydroxyl groups of the CA chains and, afterward, CA chains deacetylation. There were two clear stages of thermal degradation for this material, as previously mentioned: The first step (300–380 °C/Ts 354.7 °C) with a mass loss of 63.3% agrees with the main thermal decomposition stage and could be assigned to CA chain decomposition owing to the interruption of glycosidic bonds [8]. The second (380 to 460 °C/Ts 452 °C) with a mass loss of 20.7% was assigned to CA carbonization, resulting in full degradation of the organic part of the film [8].

Thus, the residual hits 9% owing to the slight contribution of oxides in the FAC-based composition. Additionally, Kuila et al. presented that annealing of GO at 450 °C or above is equivalent to a chemical reduction process by hydrazine monohydrate at 80 °C shadowed by heating at 200 °C [24]. The previous studies state the thermal decomposition of the TiO_2_ nanoparticles that culminated in water evaporation, then the evaporation of inorganic material within a sample, and the evaporation of unreacted organic impurities. It reported a total weight loss of 3.9% from room temperature to 800 °C [8]. This points to the slight loss owing to mineral oxides, especially with a minor contribution. In addition, the attractive interactions between the film ingredients such as hydrogen bonding might play a stabilization role [25]. This behavior is different from that detected for pure CA films [8]. On the other hand, the more compact structure (films) offers a larger number of polymeric chains and their electrostatic interactions with inserted oxides [23].

### 3.5. Optical Study

Figure 7 shows the optical behavior of CA, Cr_2_O_3_@CA, TiO_2_@CA, Cr_2_O_3_/TiO_2_@CA, and Cr_2_O_3_/TiO_2_/GO@CA films. The absorption behavior demonstrates the fact the oxides insertion causes clear deviations in the absorption pattern. These tightly studied insertions lead to clear increases in CA peak intensity with the appearance of new peaks related to the inserted oxides. Thus, it is significant that all oxy composites show a higher position than CA. The peak positioned at 239 nm is a point to n → π* transitions. There is a minor shift with oxides merging, which reflects the good dispersion of oxygenated ingredients via the polymeric medium. Additionally, the CA polymer spectrum was almost flat after 300 nm [26]. The TiO_2_ NPs showed a near-UV absorbance peak at 345 nm, while aggregated TiO_2_ NPs were shown at 395 nm. The band gap energy of the TiO_2_ NPs was valued above 3.503 eV.

Chromium III oxide existence is confirmed by a rounded peak at 405 nm (introducing a band gap of 3.08, and 3.38 eV in Cr_2_O_3_ NP, and Cr_2_O_3_, respectively). Regarding these facts and tabulated data, NC film formation causes a significant reduction in band gaps in comparison with pure mineral oxides band gaps.

The absorption coefficients (*α*) are detected via Beer–Lambert’s formula:(1)$$\alpha \left(\lambda \right)=\frac{2.303\;A}{d}$$
where *A* is absorbance and *d* is the film’s thickness. As displayed in Figure 7, *α* is measured as a function of (photon energy *hν*) and it is clear that the absorption edge is repositioned along the *x*-axis with varying mixed oxides amount and types. It started from 4.9 eV for CA, and then it minimized to 2.2 eV for TiO_2_@CA. Additionally, the band gap could be calculated using the next equation:(2)$$\alpha h\nu =A{\left(h\nu -Eg\right)}^{m}$$
where *hν* is the photon energy, *Eg* is the band gap, and *A* refers to the band tailing parameter. The power (*m*) refers to the type of transition; where (direct if *m* = 0.5, and indirect if *m* = 2). The diminishing of band gap is accompanied by the raising in crystallographic ordering, this is explained by the effect of mixed oxides insertions through the CA metrics.

The refractive index (*n*) could be found via Dimitrov and Sakka’s equation as a function of indirect energy band gap as follows:(3)$$\frac{n^{2}-1}{n^{2}+1}=1-\sqrt{\frac{Eg_{i}}{20}}$$

The refractive index parameter increases from 1.76 for CA to 2.14 for TiO_2_@CA film. As a result, the composite formation merging tactic is crucial in refractive index alteration, as demonstrated in Table 2. The refractive index does not provide distinctive values for each component because its measurement depends strongly on the thickness of the sample and also the preparation conditions of the materials. It was reported that CA might possess a refractive index of 1.5. Thus, the increase in the refractive index could reflect the increase in polymeric density due to the additional metal oxides.

### 3.6. Cell Viability (Normal)

Normal lung cell (A138) growth examination as a function of Cr_2_O_3_/TiO_2_/GO@CA NC film concentration is displayed in Figure 8. In other words, the size and distribution of the cells through and over the films were observed. The usage of 4.9 µg/mL shows a cell viability percentage of 93.46%, while 156.25 µg/mL hits 86.35%. On the subject of the usage of the highest concentrations of 5000 and 10,000 µg/mL, the cell viability % declined to 61.35 and 55.19%, respectively. The viability results point to the minor toxicity of Cr_2_O_3_/TiO_2_/GO@CA with a definite threshold. The cell viability was higher than 50% at a very high concentration of (10,000 µg/mL), which indicates the high biocompatibility of these scaffolds to be used for dressing applications. Khalid et al., 2017, have reported the successful merging between TiO_2_ NPs into cellulose [16]. The NC bandage displayed good healing with 71% wound contraction. Histopathological evidence such as the re-epithelization refers to the healing progress [16]. Prakasha et al., 2020, have stated GO/TiO_2_ NC has clear antibacterial activity in the presence of light. This is explained by the photocatalytic activity of NCs, besides the release of ROS in light [17]. Additionally, the GO/TiO_2_ NC displays significant biocompatible and hemocompatibility in this study, also displaying potential in wound healing applications [17]. Accordingly, NC films based on these ingredients have various biomedical potentials. Regarding the previous studies [27,28,29,30,31,32], as shown in Table 3, it was reported that A. Ullah studied the modification of CA for wound healing utilization. The results showed that the incorporation of manuka honey into CA nanofiber led to the formation of nano-composites that were able to inhibit the colonization of bacterial cells over the surface of wounds. Furthermore, the high porosity of the fabricated scaffolds might introduce a transfer of moisture through the wound [31].

## 4. Conclusions

Herein, cellulose acetate-based films showed biocompatibility behavior. The chromium oxide/titanium dioxide/graphene oxide integrated cellulose acetate (CA) films were fabricated. SEM micrographs of Cr_2_O_3_/TiO_2_/GO@CA film revealed Cr_2_O_3_ and TiO_2_ ingredients on a nano-scale. The EDX analysis resultant data confirmed the existence of C, N, O, Ti, and Cr elements. The TiO_2_@CA showed the lowest contact angle with 30°, besides introducing the highest refractive index with 2.14. Likewise, normal lung cells (A138) growth examinations were introduced at 93.46% with exposure of 4.9 µg/mL. The resulting data promoted the usage of the tested NCs for wound healing hastening.

## Figures and Tables

**Figure 1 polymers-15-00485-f001:**
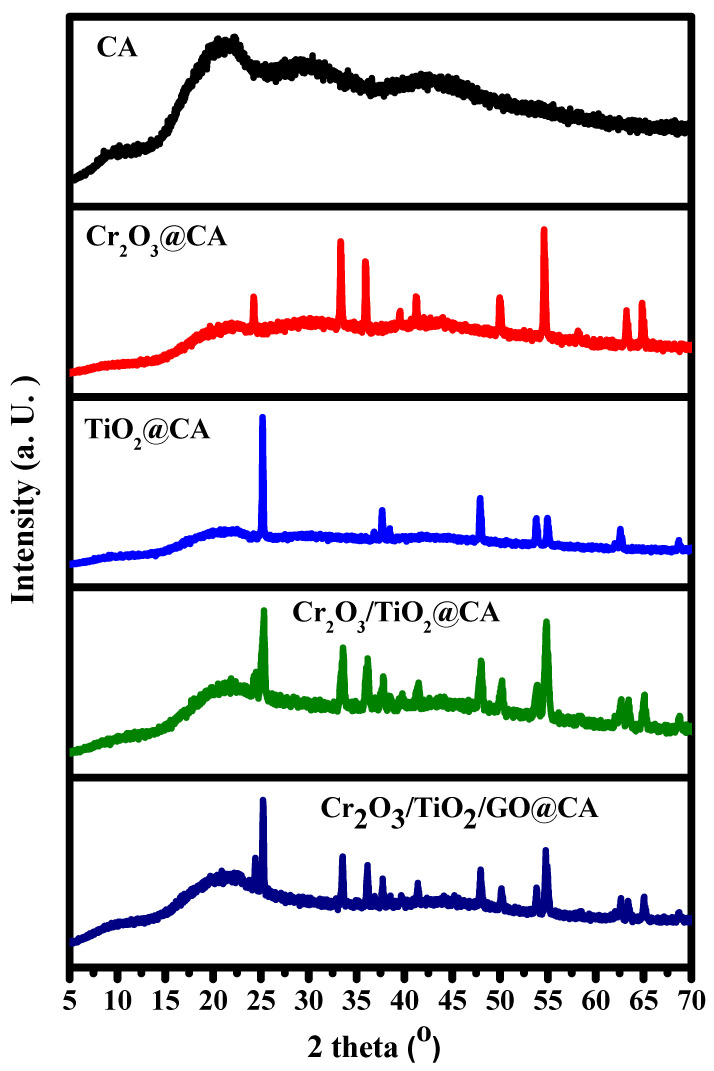
XRD of the CA-based films containing TiO_2_, Cr_2_O_3_, and graphene oxide; CA, Cr_2_O_3_@CA, TiO_2_@CA, Cr_2_O_3_/TiO_2_@CA, and Cr_2_O_3_/TiO_2_/GO@CA.

**Figure 2 polymers-15-00485-f002:**
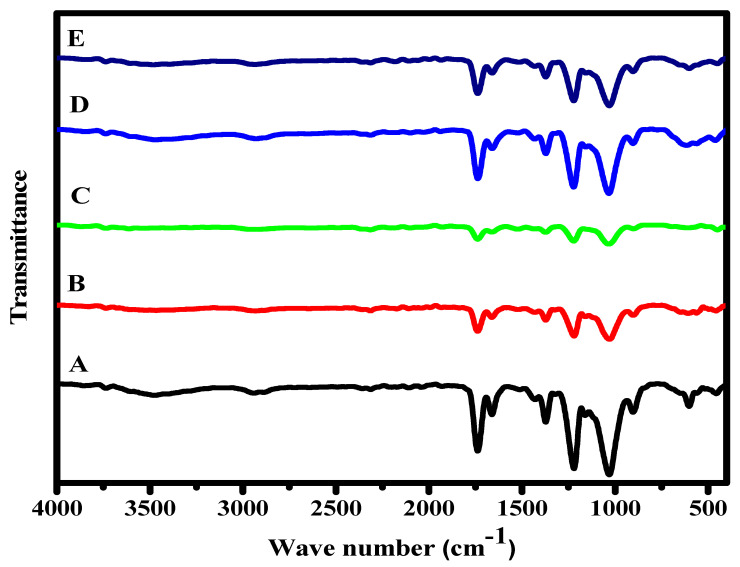
FTIR of CA-based nano-composites; (A) CA, (B) Cr_2_O_3_@CA, (C) TiO_2_@CA, (D) Cr_2_O_3_/TiO_2_@CA, and (E) Cr_2_O_3_/TiO_2_/GO@CA.

**Figure 3 polymers-15-00485-f003:**
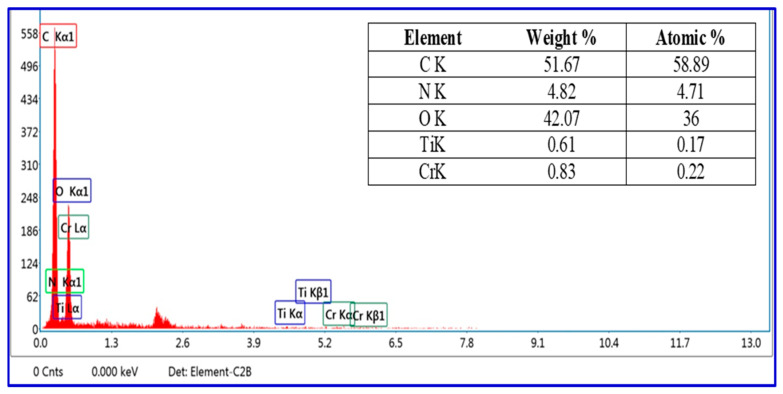
EDX of Cr_2_O_3_/TiO_2_/GO@CA. The data of EDX analysis of Cr_2_O_3_/TiO_2_/GO@CA film (inset).

**Figure 4 polymers-15-00485-f004:**
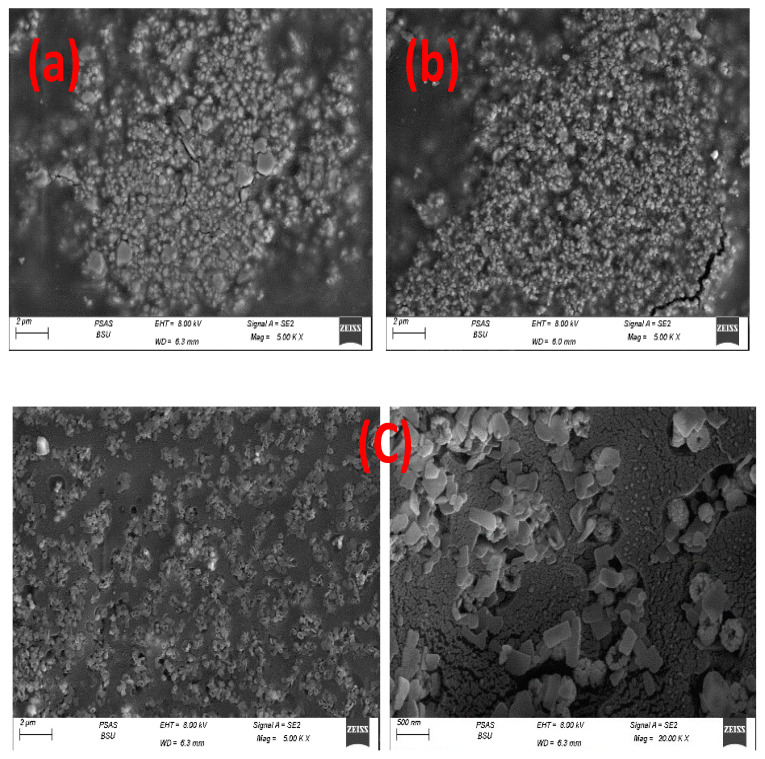
FESEM micrographs of CA based films; where (**a**) Cr_2_O_3_@CA, (**b**) TiO_2_/Cr_2_O_3_@CA, (**c**) Cr_2_O_3_/TiO_2_/GO@CA.

**Figure 5 polymers-15-00485-f005:**
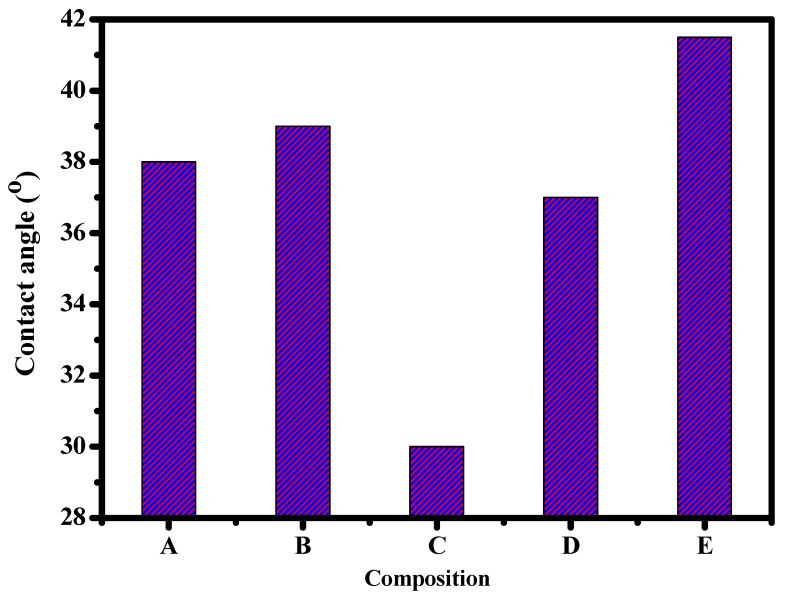
Contact angle of CA-based films doped with Cr_2_O_3_, TiO_2_ and GO; (A) CA, (B) Cr_2_O_3_@CA, (C) TiO_2_@CA, (D) Cr_2_O_3_/TiO_2_@CA, and (E) Cr_2_O_3_/TiO_2_/GO@CA.

**Figure 6 polymers-15-00485-f006:**
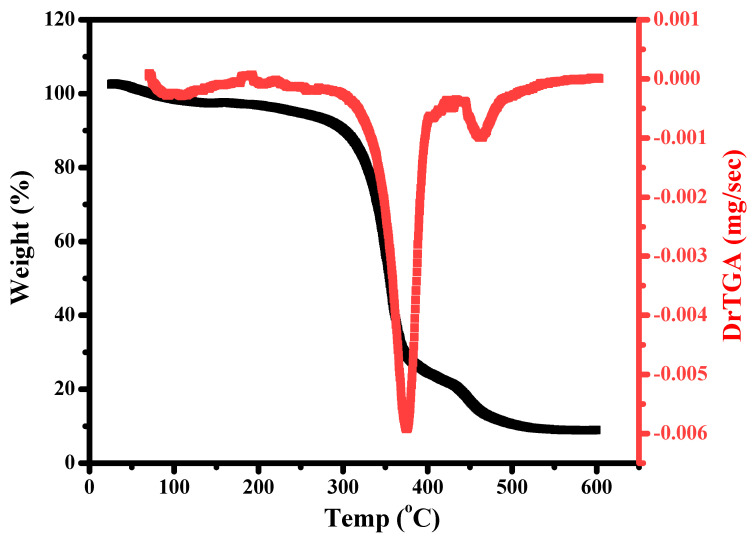
TGA of Cr_2_O_3_/TiO_2_/GO@CA cast films.

**Figure 7 polymers-15-00485-f007:**
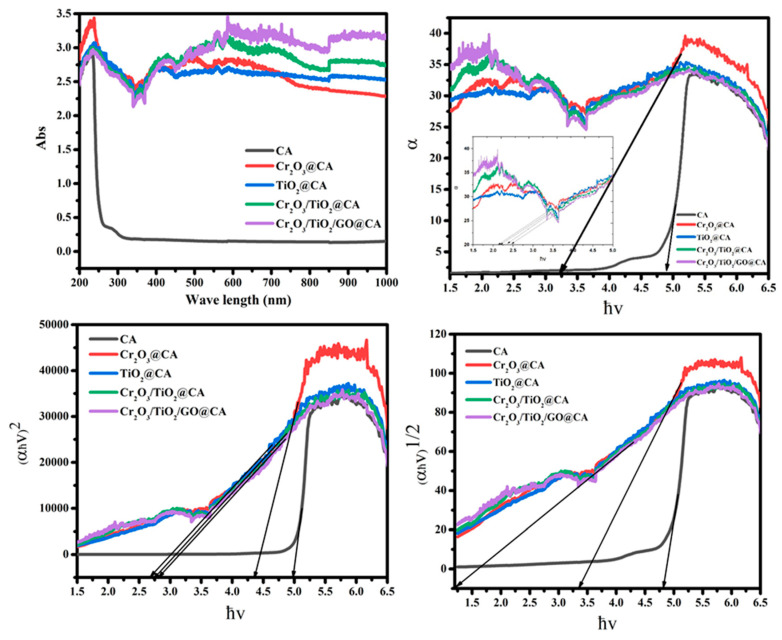
The optical behavior of CA-based nano-composites.

**Figure 8 polymers-15-00485-f008:**
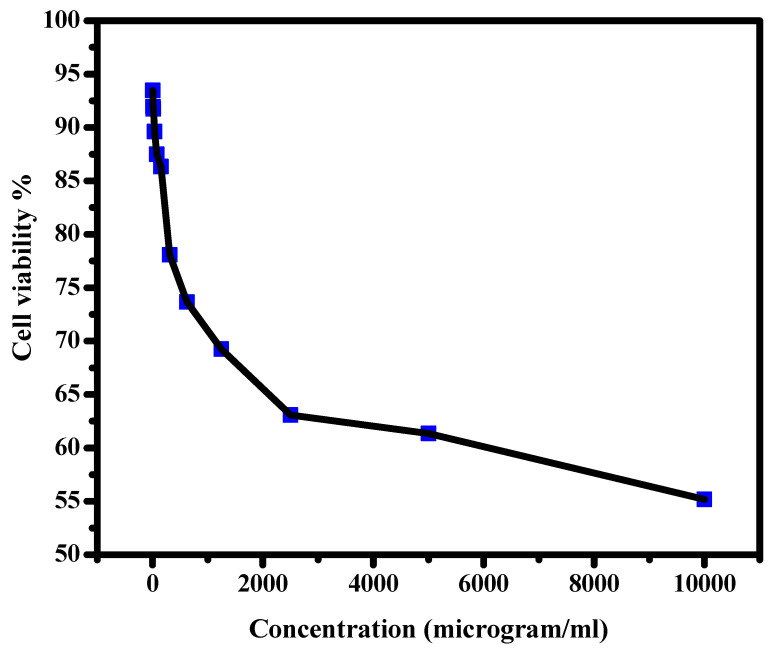
Cell viability of Cr_2_O_3_/TiO_2_/GO@CA against normal lung cells (IC_50_ > 10,000 microgram/mL).

**Table 1 polymers-15-00485-t001:** The chemical composition of the studied nano-film scaffolds.

Composition	Oxides			Polymeric Constituent	Amount (g)			Polymeric Concentration
CA				CA	0	0	0	10%
Cr_2_O_3_@CA	Cr_2_O_3_			CA	0.25	0	0	10%
TiO_2_@CA		TiO_2_		CA	0	0.25	0	10%
Cr_2_O_3_/TiO_2_@CA	Cr_2_O_3_	TiO_2_		CA	0.125	0.125	0	10%
Cr_2_O_3_/TiO_2_/GO@CA	Cr_2_O_3_	TiO_2_	GO	CA	0.1	0.1	0.05	10%

**Table 2 polymers-15-00485-t002:** Optical properties of CA-based NCs, including absorption edge, direct and indirect band gaps, and refractive index (*n*).

Composition	Absorption Edge (eV)	Band Gap (eV)		N
		Direct	Indirect	
CA	4.9	4.9	4.9	1.76
Cr_2_O_3_@CA	3.2	3.4	4.4	1.8
TiO_2_@CA	2.2	1.0	2.65	2.14
Cr_2_O_3_/TiO_2_@CA	2.4	1.1	2.7	2.12
Cr_2_O_3_/TiO_2_/GO@CA	2.5	1.2	2.75	2.09

**Table 3 polymers-15-00485-t003:** Comparison between the previous results for similar compositions based on CA modified with different additives for medical applications.

Composition	Application	Main Results	Ref.
Cellulose nanofiber (CNF)/Ag nanoparticles	Wound healing	The AgNPs show antioxidant and antimicrobial activity. The anti-diabetic AgNPs were observed to be 56% and 61%.	[27]
CNF/Doxorubicin	Tumor/infection-induced wound healing	High effectiveness against both tumor cells and bacterial invasion.	[28]
CA nanofiber/GO/TiO_2_/Curcumin	Wound healing	High hemocompatibility and promoted cell proliferation and migration.	[29]
Lignin/CuO/CA nanofiber	Wound healing	The Cu/CA showed faster release (80%) of copper ions within 24 h.	[30]
Manuka honey/CA nanofiber	Wound healing	High efficacy to prevent bacterial growth, high porosity suitable for wound breathability, and high cytocompatibility.	[31]
Ag-sulfadiazine/CA nanofiber	Wound healing	Appreciable antimicrobial activity against E. Coli and Bacillus Subtilis.	[32]

## Data Availability

The raw/processed data generated in this work are available upon request from the corresponding author.

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
