# Peer review of "Titanium Dioxide/Chromium Oxide/Graphene Oxide Doped into Cellulose Acetate for Medical Applications"

_polymers, 2023, doi:10.3390/polym15030485_

Round 1
Reviewer 1 Report (Previous Reviewer 1)
The revised form of this manuscript is appropriate for the publication.
Reviewer 2 Report (Previous Reviewer 2)
The authors addressed most of the comments and the revised version can be accepted.
This manuscript is a resubmission of an earlier submission. The following is a list of the peer review reports and author responses from that submission.
Round 1
Reviewer 1 Report
The manuscript was about the preparation of cellulose acetate films incorporated w,th titanium oxide, chromium oxide and graphene oxide. There exists many major problems throughout the study about both experimental and results sections.
The preparation of the samples was not explained in detail. For example, the reason for the selected amounts of the ingredients was undefined. The washing step after the synthesis was not existed in text. The methods of optical properties was not fully explained, the information was not spesified. The experimental methods of FTIR and contact angle measurements were not existed in the experimental part, although the results were presented of these analyses. In the FTIR spectrum, the bands were not clearly observable. The OH bands should be presented more precisely if the main interaction depends on these groups. The incorporation of GO was not proved by any of the characterization results since there was no control group for the understanding of the incorporation. Also in the TGA analyses, there exists any control groups. The refreactive index values were given and there was any explanation about the compatilibty with the literature. For the MTT tests, the minor toxicity was declared. In the conclusion, the samples were said to be compatible for wound dressing applications.Â
Reviewer 2 Report
The authors reported Chromium oxide/titanium dioxide/ graphene oxide/cellulose acetate cast films for Wound Healing. The submission requires major revision taking into account the following points:-
 1.     The title should be revised to be short, precise, and informative. Redundant words such as ‘ Novel’; ‘encapsulated’; ‘cast films’; and ‘Applications’ should be removed.
2.     The role of each component in the composite, Cr2O3/ TiO2/GO@CA films, should be discussed.
3.     Figure 1 should be revised to be comparable. The simulated XRD pattern of the materials should be also included.
4.     The mass ratio of the components should be added. Based on Table 1, the film contains 10 % polymer. While, TGA shows only 9% of the oxide.
5.      The title shows ‘wound healing. While, there is only one Figure showing Cell viability of Cr2O3/ TiO2/GO@CA against normal lung cells. Please, revise it accordingly.
6.      The biological activity of all materials; Cr2O3@CA, TiO2@CA, Cr2O3/ TiO2@CA, and 253 Cr2O3/ TiO2/GO@CA films should be tested.
7.      The intrinsic toxicity of the heavy metal should be considered.
8.      The text should be revised to improve the study's clarity with the proper References. Wound healing using cellulose and GO was reported in Ref. https://doi.org/10.1016/j.ijbiomac.2020.07.160; https://doi.org/10.3390/ijms23105405; https://doi.org/10.3390/ph14121215; https://doi.org/10.1016/j.rvsc.2021.05.013.
9.      A comparison with previously reported materials should be discussed and summarized in a table.
10.   The language should be revised and typos should be corrected.
Â
Â
Â
Â
Â